# Evaluation of the Molecular Mechanism of Chlorogenic Acid in the Treatment of Pulmonary Arterial Hypertension Based on Analysis Network Pharmacology and Molecular Docking

Jovito Cesar Santos-Álvarez [1,†], Juan Manuel Velázquez-Enríquez [1,*,†] and Rafael Baltiérrez-Hoyos [1,2,*]

[1] Laboratorio de Fibrosis y Cáncer, Facultad de Medicina y Cirugía, Universidad Autónoma Benito Juárez de Oaxaca, Ex Hacienda de Aguilera S/N, Sur, San Felipe del Agua, Oaxaca 68020, Mexico; jovitocesarsa@cecad-uabjo.mx

[2] CONAHCYT-Facultad de Medicina y Cirugía, Universidad Autónoma Benito Juárez de Oaxaca, Ex Hacienda de Aguilera S/N, Sur, San Felipe del Agua, Oaxaca 68020, Mexico

[*] Correspondence: juanmanuelvela_enriquez@cecad-uabjo.mx (J.M.V.-E.); rbaltierrezho@conacyt.mx (R.B.-H.)

[†] These authors contributed equally to this work.

**Abstract:** Background: Pulmonary arterial hypertension (PAH) is a serious disease characterized by increased pressure in the pulmonary arteries, which can lead to heart failure and death. Chlorogenic acid (CGA) is a natural compound present in several foods and medicinal plants and has been described to exert a therapeutic effect in various diseases. However, its potential therapeutic effect on PAH remains undeciphered. In this study, the potential of CGA for the treatment of PAH was investigated using network pharmacology analysis and molecular docking. Methods: Potential CGA targets were obtained from the SwissTargetPrediction and GeneCards databases. Moreover, potential PAH targets were collected from the GeneCards and DisGNET databases. Then, common targets were selected, and a protein-protein network (PPI) was constructed between common CGA and PAH targets using the STRING database. The common hub targets were selected, and GO enrichment analysis was performed via KEGG using the DAVID 6.8 database. Additionally, molecular docking analysis was performed to investigate the interaction between CGA and these potential therapeutic targets. Results: We obtained 168 potential targets for CGA and 5779 potential targets associated with PAH. Among them, 133 were common to both CGA and PAH. The main hub targets identified through PPI network analysis were TP53, HIF1A, CASP3, IL1B, JUN, MMP9, CCL2, VEGFA, SRC, IKBKB, MMP2, CASP8, NOS3, MMP1, and CASP1. KEGG pathway analysis showed that these hub targets are associated with pathways such as lipid and atherosclerosis, fluid shear stress and atherosclerosis, and the IL-17 signaling pathway. In addition, the molecular docking results showed a high binding affinity between CGA and the 15 hub PAH-associated targets, further supporting its therapeutic potential. Conclusions: This study provides preliminary evidence on the underlying molecular mechanism of CGA in the treatment of PAH. The findings suggest that CGA could be a promising option for the development of new PAH drugs.

**Keywords:** chlorogenic acid; pulmonary arterial hypertension; pulmonary disease; network pharmacology; molecular docking; drug discovery

## 1. Introduction

Pulmonary arterial hypertension (PAH) is one of the five groups of pulmonary hypertension (PH) and is characterized as severe and progressive, with elevated pulmonary vascular resistance and pulmonary artery pressure [1]. PAH is a complex and multifaceted condition characterized by several hallmark features, such as inflammation, impaired angiogenesis, metabolic alterations, and the deposition of extravascular collagen [2–4]. The microenvironment composition plays a crucial role in the development of PAH, involving

various factors such as enzymes, transcription factors, proteins, chemokines, cytokines, hypoxia, and oxidative stress [5,6].

Food and Drug Administration (FDA) approved monotherapy treatments for PAH primarily target vasodilation, endothelial dysfunction, and antiproliferative mediators and have been effective in improving the symptoms and survival of PAH patients. However, despite these treatments, mortality rates among high-risk patients remain high, with a three-year survival rate and a median age of 56 [7,8]. More researchers are searching for treatments focused on herbal medicine and their bioactive components effective in different pathologies, as well as in the treatment of PAH.

Chlorogenic acid (CGA) is a phenolic compound that is found in a wide variety of plants, including berries, tea, grapes, apples, cocoa, pears, carrots, artichokes, potatoes, aubergines, kiwis, tomatoes, and coffee [9,10]. Coffee is one of the most internationally consumed beverages [11] and has been attributed to exhibit benefits in Alzheimer's disease (AD), nonalcoholic fatty liver disease (NAFLD), some cancers [12], and cardiovascular disease (CVD). CGA has been attributed different biological functions, such as lowering blood glucose, anti-inflammatory effects, antioxidant properties, and antibacterial, antiviral, and anticarcinogenic activities [9,13–15], and importantly, it has been shown to reduce hypertension [16]. Recent research has suggested that CGA may have a potential therapeutic effect on various types of lung diseases including PAH [17–19]. It has been suggested that CGA prevents the oxidative, fibrotic, and inflammatory effects of paraquat (PQ)-induced lung toxicity by enhancing antioxidant enzymes in murine models [17]. In addition, CGA was shown to reduce PQ-induced lung epithelial cell (AEC) apoptosis by preventing caspase 3 and caspase 9 cleavage and cytochrome c release from mitochondria to the cytoplasm, as well as reducing reactive oxygen species (ROS) production by increasing the levels of the NF-E2-related factor 2 (Nrf2), superoxide dismutase 2 (SOD2), and glutathione [20]. Furthermore, CGA has been shown to inhibit the proliferation of human lung cancer A549 cell lines by inhibiting the binding of annexin A2 to the p50 subunit, resulting in the regulation of the expression of the antiapoptotic genes cIAP1 and cIAP2 of the NF-κB signaling pathway, resulting in a significant reduction in the proliferation of these tumor cell lines [18]. Additionally, available evidence suggests that CGA alleviates bleomycin-induced pulmonary fibrosis in mice by significantly improving lung inflammation and the degree of fibrosis through the inhibition of endoplasmic reticulum stress, autophagy, and epithelial–mesenchymal transition (EMT) [19,21]. In relation to PAH, studies have suggested that CGA is capable of inhibiting hypoxia-induced pulmonary artery smooth muscle cell (PASMC) proliferation, one of the cellular processes closely related to vascular remodeling in PAH. These inhibitory effects were associated with reduction in alpha hypoxia inducible factor 1 subunit alpha (HIF-1$\alpha$) expression, G1 cell cycle arrest, and the negative regulation of cell cycle proteins. Trials conducted in a murine model of monocrotaline-induced PAH in rats showed that CGA alleviates PAH by reducing intrapulmonary arterial hyperplasia [22]. However, despite the potential benefits of CGA in the treatment of lung diseases, such as PAH, the underlying molecular mechanism remains largely unexplored. Although the beneficial effects of CGA have been observed, a deeper understanding of how this molecule exerts its therapeutic effects is crucial.

An essential aspect of disease treatment is focusing on drugs that can target multiple proteins involved in a disease's characteristics. Network pharmacology is a valuable tool for designing such drugs, and it involves constructing multiple networks to understand the interactions among gene targets, diseases, drugs, and related Kyoto Encyclopedia of Genes and Genomes (KEGG) pathways based on systems biology, computational biology, and omics theory [23,24]. Additionally, molecular docking allows for the calculation of binding energies between receptors and ligands, predicting reasonable binding modes [25,26].

The aim of this study was to investigate the potential targets and mechanisms of action of CGA for the treatment of PAH via network pharmacology and molecular docking. This approach is expected to identify new therapeutic targets for PAH and provide a molecular basis for the use of CGA in the treatment of the disease.

## 2. Materials and Methods

### 2.1. Obtaining CGA-Related Target Genes

The canonical molecular structure and SMILES file of CGA were obtained from the PubChem database (https://pubchem.ncbi.nlm.nih.gov/, accessed on 26 April 2023) [27]. Subsequently, CGA target genes were obtained with the help of the SwissTargetPrediction (http://www.swisstargetprediction.ch/, accessed on 26 April 2023) and GeneCards (https://www.genecards.org/, accessed on 26 April 2023) databases [28,29]. All CGA-associated targets were transformed into the UniProt database in the protein identification format using the Retrieve/ID mapping tool (www.uniprot.org, accessed on 26 April 2023) [30,31]. The selected targets were then pooled, and duplicate values were removed from the study to obtain the target genes for CGA.

### 2.2. Obtaining PAH-Related Target Genes

Using the keyword "Pulmonary arterial hypertension" as the search term, PAH-associated target genes were selected from the GeneCards database (https://www.genecards.org/, accessed on 26 April 2023) and the DisGeNET database (https://www.disgenet.org/, accessed on 26 April 2023) [29,32]. All PAH-associated targets were transformed into the UniProt database in the protein identification format using the Retrieve/ID mapping tool (www.uniprot.org, accessed on 26 April 2023 [30,31]. The selected targets were pooled, and duplicate values were removed from the study to obtain the target genes for PAH.

### 2.3. Obtaining Potential Common PAH and CGA Targets

To obtain common potential targets of PAH and CGA, the potential targets of PAH and CGA were imported into a Venn diagram using the online platform Venny 2.1 (https://bioinfogp.cnb.csic.es/tools/venny/index.html, accessed on 26 April 2023).

### 2.4. Protein-Protein Interaction (PPI) Network Construction and Detection of Hub Targets

To explore the interaction networks of CGA target genes for PAH, an interaction network was constructed using the Search tool for retrieval of interacting genes/proteins (STRING) version 11.5 (https://string-db.org/, accessed on 27 April 2023), and the PPI network was constructed with a minimum required interaction score of >0.4 [33]. Visualization of the PPI network was performed using Cytoscape v.3.8.2 software, and then, Cytoscape's cytoHubba add-on was used to explore the hub target genes of the PPI network using the maximum clique centrality (MCC) method [34].

### 2.5. Gene Ontology (GO) Enrichment and KEGG Pathway Enrichment Analysis

The Database for Annotation, Visualization, and Integrated Discovery (DAVID) online tool (https://david.ncifcrf.gov/, accessed on 27 April 2023) was used for the enrichment analysis of CGA target genes against IPF, including GO and KEGG pathway analyses [35]. The GO analysis includes the analysis of three main aspects: cellular component (CC), molecular function (MF), and biological process (BP). Gene set enrichment results with $p < 0.05$ were considered statistically significant.

### 2.6. Molecular Docking

The structure of CGA was obtained from the PudChem database (https://pubchem.ncbi.nlm.nih.gov/compound/1794427, accessed on 28 April 2023) [27]. For modeling, from crystallized structures present in the Protein Data Bank database (https://www.rcsb.org/, accessed on 30 March 2023) [36], the most efficient protein crystal structure (human protein, most complete structure, and lowest resolution value) was selected. Ligands and water molecules were removed, and appropriate hydrogens and charges were added using Autodock and Discovery Studio software [37]. Molecular docking between the refined ingredients and the protein was performed using AutoDock Vina, CB-Dock2 (https://cadd.labshare.cn/cb-dock2/php/index.php, accessed on 7 May 2023) [37–39], and LIGPLOT software was used to visualize the hydrogen bridges and distances to CGA [40].

### 3. Results

*3.1. Screening of Potential PAH Targets for CGA*

Potential targets for PAH and CGA were obtained from various databases. After querying and analyzing the SwissTargetPrediction and GeneCards databases, 100 and 77 potential CGA-associated targets were identified, respectively (Tables S1 and S2). After the integration and removal of duplicate values, 168 potential CGA-associated targets were retrieved. At the same time, after querying and analyzing the GeneCards and DisGeNET databases, 5744 and 373 potential PAH-associated targets were identified, respectively (Tables S3 and S4), and 5779 potential PAH-associated targets were recovered after integrating and removing duplicate values. Subsequently, the potential targets of CGA and PAH were analyzed using a Venn diagram to identify overlapping targets, and 133 common targets (2.3% of the total) were retrieved (Figure 1, Table S5).

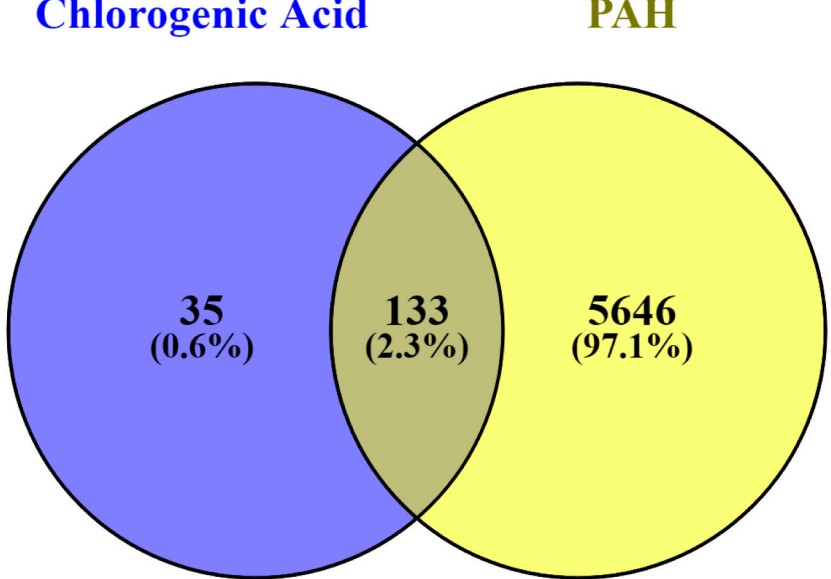

**Figure 1.** Venn diagram showing the common targets between CGA and PAH. The blue and yellow colors represent the targets related to CGA and PAH, respectively. The intersection represents the common targets between CGA and PAH.

*3.2. PPI Network Construction and Hub Target Analysis*

The 133 common targets were loaded into the STRING v11.5 database to construct the PPI network and reveal the correlation between the potential targets. The results show that the PPI network contains 91 nodes and 412 edges with an average node degree of 9.05 and a clustering coefficient of 0.526. The expected number of edges was 154, which was much smaller than the actual number of edges found, and the *p* value of PPI enrichment was $<1.0 \times 10^{-16}$ (Figure 2).

Subsequently, using the cytoHubba add-on of Cytoscape software, the PPI network was examined to identify the top 15 hub targets of the PPI network using the MCC algorithm, and a network diagram of the hub targets of CGA and PAH was constructed. The results showed that the top 15 hub targets were tumor suppressor p53 (TP53), hypoxia-inducible factor 1-alpha (HIF1A), caspase-3 (CASP3), interleukin-1 beta (IL1B), transcription factor Jun (JUN), matrix metalloproteinase-9 (MMP9), C-C motif chemokine 2 (CCL2), vascular endothelial growth factor A (VEGFA), proto-oncogene tyrosine-protein kinase Src (SRC), inhibitor of nuclear factor kappa-B kinase subunit beta (IKBKB), matrix metalloproteinase-2 (MMP2), caspase-8 (CASP8), nitric oxide synthase, endothelial (NOS3), matrix metalloproteinase-1 (MMP1), and caspase-1 (CASP1) (Figure 3).

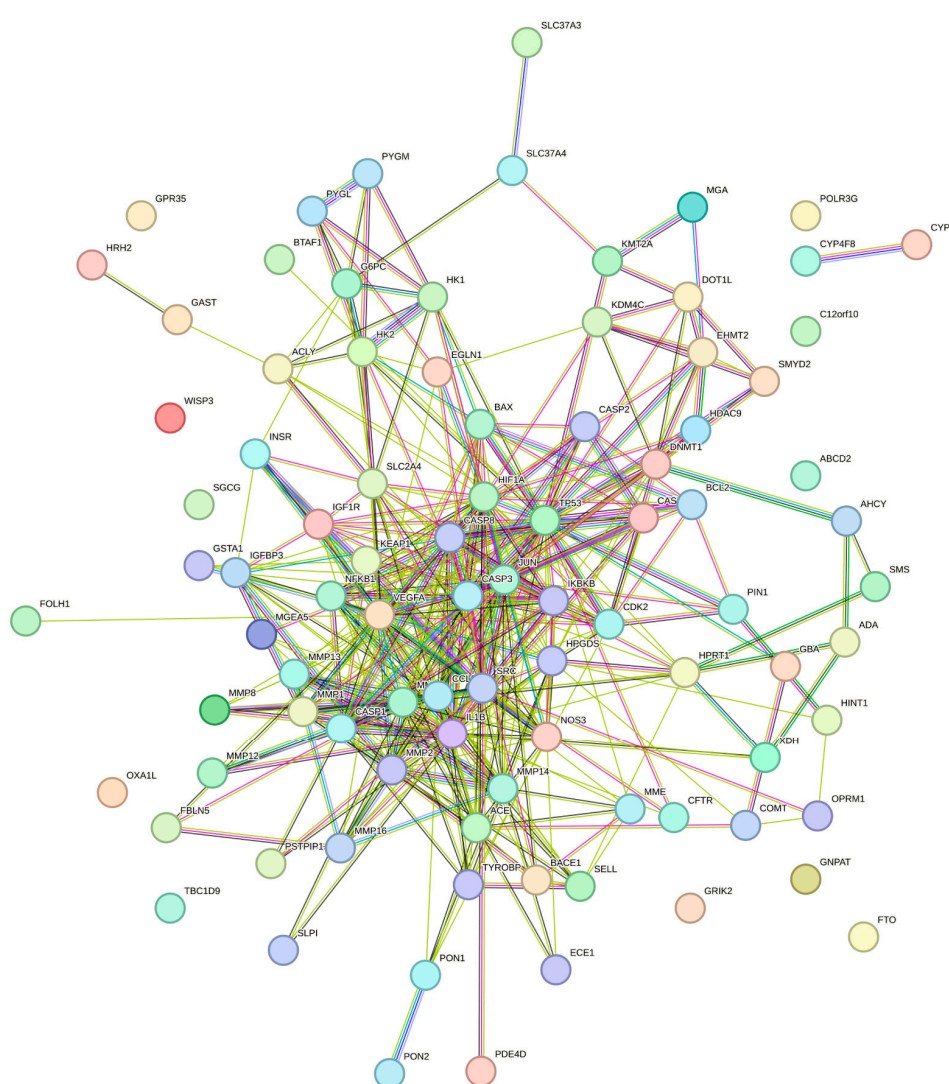

**Figure 2.** PPI network of common targets between CGA and PAH.

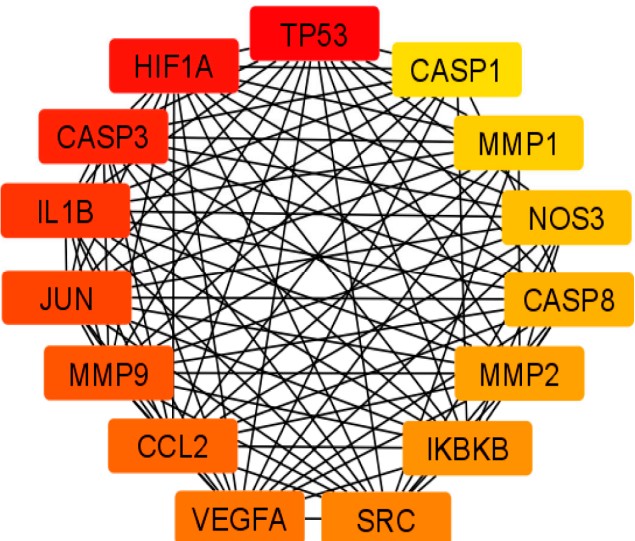

**Figure 3.** Top 15 hub targets in the PPI network. The 15 hub targets with the highest degree of PPI network connectivity were identified with the MCC method using cytoHubba, red colour represents a higher degree, and yellow colour represents a lower degree.

*3.3. GO and KEGG Pathway Enrichment Analyses*

To understand and illustrate the molecular mechanism by which CGA affects PAH, GO enrichment and KEGG pathway analyses were performed using the online tool DAVID 6.8. The GO enrichment analysis includes three main branches: BP, MF, and CC. Figure 4 illustrates the top 10 BPs, MFs, and CCs. The results show that the predicted hub targets for CGA in PAH were enriched in BP, such as angiogenesis, the positive regulation of apoptotic process, apoptotic process, the positive regulation of vascular smooth muscle cell proliferation, the positive regulation of transcription from RNA polymerase II promoter in response to hypoxia, proteolysis, the negative regulation of apoptotic process, response to xenobiotic stimulus, cellular response to UV-A, and cellular response to hypoxia. Similarly, the results obtained suggest that these central targets were associated with MFs, such as scaffold protein binding, cysteine-type endopeptidase activity involved in the apoptotic signaling pathway, endopeptidase activity, peptidase activity, identical protein binding, ubiquitin protein ligase binding, cysteine-type endopeptidase activity, enzyme binding, metalloendopeptidase activity, and cysteine-type endopeptidase activity involved in the apoptotic process. On the other hand, in relation to the CCs, these core targets were enriched in the caspase complex, membrane raft, death-inducing signaling complex, extra-cellular space, macromolecular complex, extracellular region, transcription factor complex, extracellular matrix, cytosol, and cytoplasm. Furthermore, the results obtained from the enrichment analysis of the KEGG pathway revealed that the predicted hub targets for CGA in PAH are related to pathways such as lipid and atherosclerosis, fluid shear stress and atherosclerosis, IL-17 signaling pathway, bladder cancer, AGE-RAGE signaling pathway in diabetic complications, Kaposi sarcoma-associated herpesvirus infection, TNF signaling pathway, pathways in cancer, human cytomegalovirus infection, and relaxin signaling pathway (Figure 4). The complete list of BPs, MFs, CCs, and KEGG pathways can be found in Table S6.

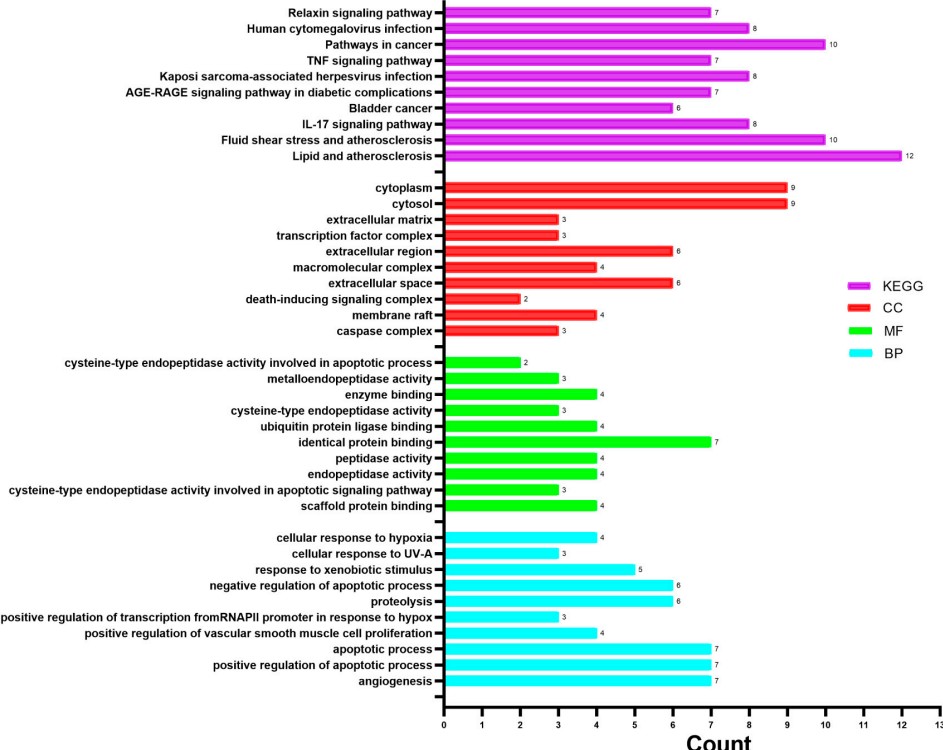

**Figure 4.** GO terms and KEGG pathway enrichment analysis of the 15 hub targets of CGA and PAH. Top 10 terms for each of the three categories of GO analysis including biological processes (BP), molecular functions (MF), and cellular components (CC) and top 10 terms for KEGG pathway enrichment analysis for hub targets.

### 3.4. Molecular Docking of Hub Targets

The 15 hub targets for PAH were screened with CGA, the binding energy of which was successfully predicted using docking analysis. The crystal structure of the selected genes was obtained and chosen with the best data from the PDB database. AutoDock Vina 1.1.2 software was used to determine the docking, and CB-Dock2 software was used to determine the size of the docking pocket. It should be noted that the lower the binding energy, the higher the estimated affinity for the binding of a protein and a ligand. The range of docking values was from −5.8 to −9.4 kcal/mol. The predicted values are listed in Table 1, and we noted that MMP2 achieved the best binding energy.

**Table 1.** Energy obtained using docking simulation.

| | Protein | ID PDB | Kcal/mol | $Å^3$ |
|---|---|---|---|---|
| 1 | TUMOR SUPPRESSOR P53 | 1KZY | −7.1 | 115 |
| 2 | HYPOXIA INDUCIBLE FACTOR 1-ALPHA | 4H6J | −5.8 | 63 |
| 3 | CASPASE-3 | 4H6J | −7.3 | 255 |
| 4 | INTERLEUKIN-1 BETA | 2NVH | −6.6 | 182 |
| 5 | C-JUN | 1JUN | −5.6 | 12 |
| 6 | MATRIX METALLOPROTEINASE-9 | 1L6J | −7.8 | 399 |
| 7 | CHEMOKINE (C-C MOTIF) LIGAND 2 | 1DOK | −5.9 | 20 |
| 8 | VASCULAR ENDOTHELIAL GROWTH FACTOR A | 1BJ1 | −5.8 | 29 |
| 9 | PROTO-ONCOGENE TYROSINE-PROTEIN KINASE SRC | 2H8H | −8.0 | 662 |
| 10 | INHIBITOR OF NUCLEAR FACTOR KAPPA-B KINASE SUBUNIT BETA | 4KIK | −8.6 | 564 |
| 11 | MATRIX METALLOPROTEINASE-2 | 7XGJ | −9.4 | 789 |
| 12 | CASPASE-8 | 3KJN | −7.3 | 154 |
| 13 | NITRIC OXIDE SYNTHASE, ENDOTHELIAL | 4D1O | −8.8 | 329 |
| 14 | MATRIX METALLOPROTEINASE-1 | 2CLT | −7.3 | 372 |
| 15 | CASPASE-1 | 2H54 | −7.2 | 152 |

We grouped the proteins in relation to the processes involved in PAH. In relation to the target proteins associated with inflammatory processes, the results showed that interleukin-1β with GCA had a score of −6.6 kcal/mol, with hydrogen bridges at the following amino acids: Pro87, Tyr90, Val3, Leu62, Lys65, Tyr68, Glu64, and Ser5. CCL2 with GCA had a score of −5.9 kcal/mol, with hydrogen bridges at the following amino acids: Arg30, Thr32, Ser33, Ser34, Lys35, Ala7, and Ala4. eNOS with GCA, had a score of 8.8 kcal/mol, with hydrogen bridges at the following amino acids: Glu432, Ser78, Gln462, Asp82, and Val465 (Figure 5).

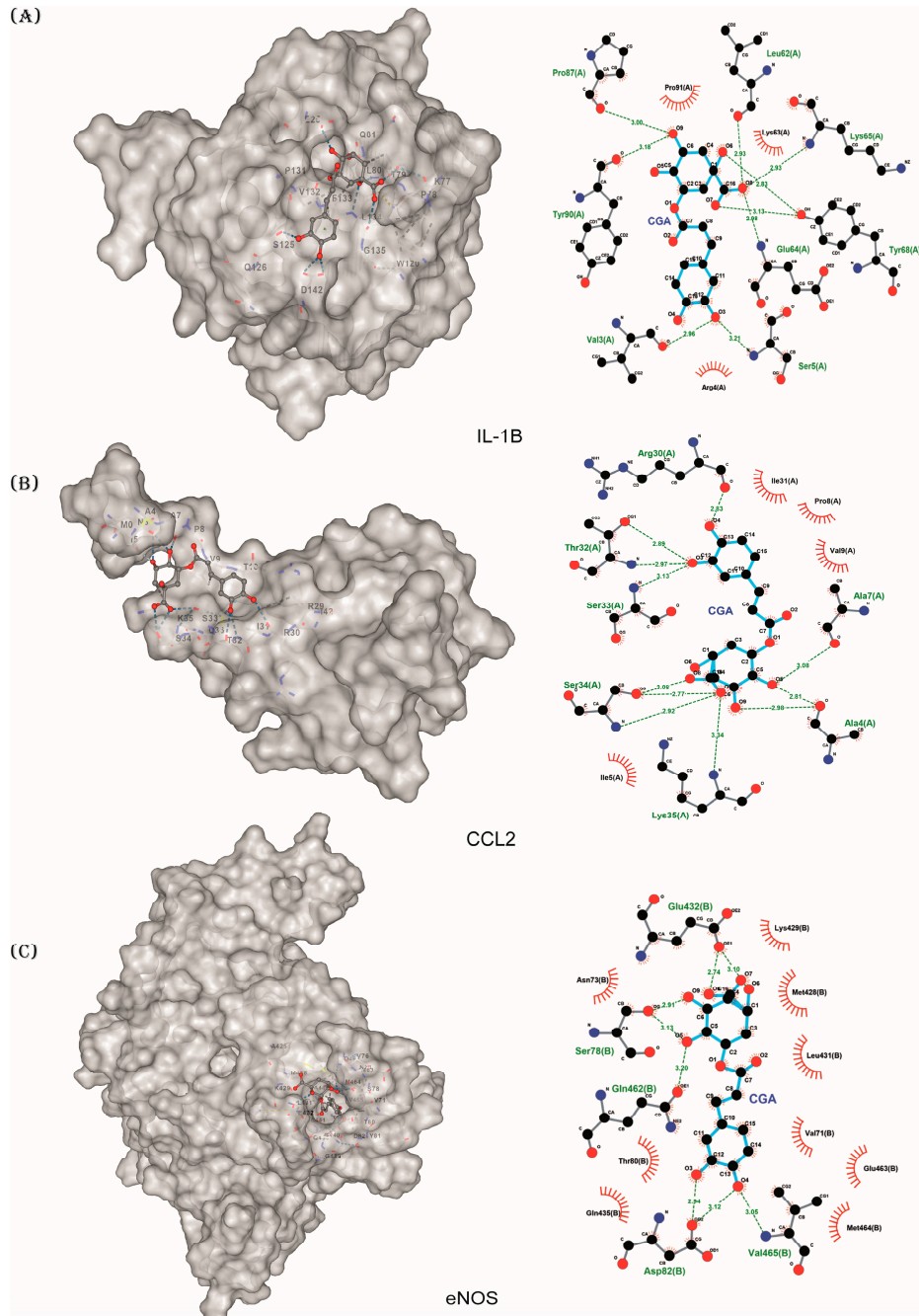

**Figure 5.** Schematic representation of molecular docking of inflammatory proteins. (**A**): IL-1B and GCA docking energy is −6.6 kcal/mol; (**B**): CCL2 and GCA docking energy is −5.9 kcal/mol; (**C**): eNOS (NOS3) and GCA docking energy is −8.8 kcal/mol.

In relation to the target proteins associated with the degradation of ECM, the results showed that MMP1 with GCA had a score of −7.3 kcal/mol, with hydrogen bridges at the following amino acids: Tyr384, Gly273, Glu314, Glu364, Thr270, Met412, and Lys413. MMP2 with GCA had a score of −9.4 kcal/mol, with hydrogen bridges at the following amino acids: Ile142, Pro141, His131, His121, His125, and Leu138. MMP9 with GCA had a score of −7.8 kcal/mol, with hydrogen bridges at the following amino acids: Thr426, Pro429, His432, Leu431, Tyr420, and Met422 (Figure 6).

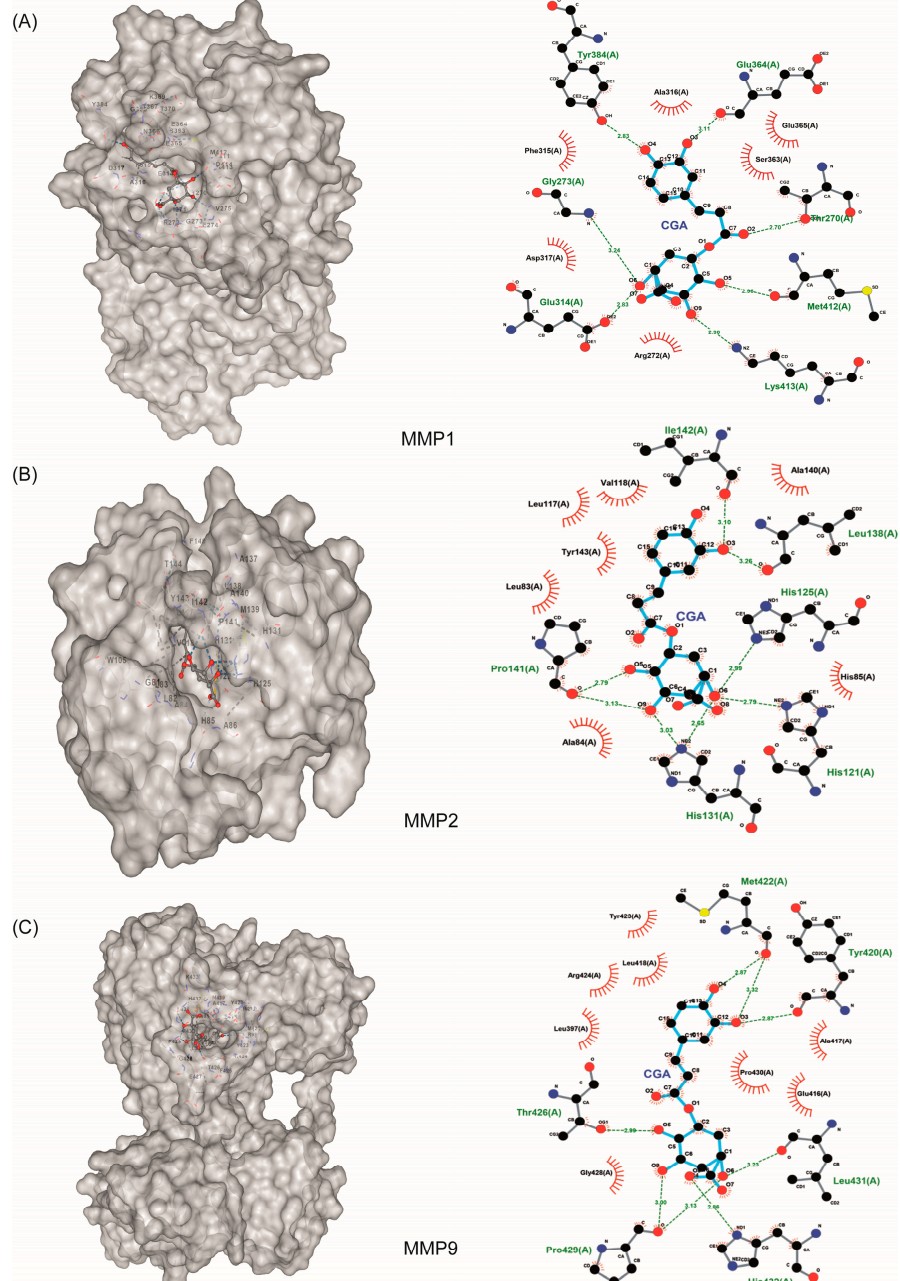

**Figure 6.** Schematic representation of molecular docking of proteins involved in ECM degradation. (**A**): MMP1 and GCA docking energy is −7.3 kcal/mol; (**B**): MMP2 and GCA docking energy is −9.4 kcal/mol; (**C**): MMP9 and GCA docking energy is −7.8 kcal/mol.

On the other hand, in relation to the target proteins associated with transcription factor processes, the results showed that JUN with GCA had a score of −5.6 kcal/mol, with hydrogen bridges at the following amino acids: Ser292, Arg302, Thr297, Glu293, Gln290, Asn291, and Lys288. HIF-A with GCA had a score of −5.8 kcal/mol, with hydrogen bridges at the following amino acids: Ser274, His291, and Thr288. TP53 with GCA had a score of −7.1 kcal/mol, with hydrogen bridges at the following amino acids: Gln1944, Asp1923, and Lys1964 (Figure 7).

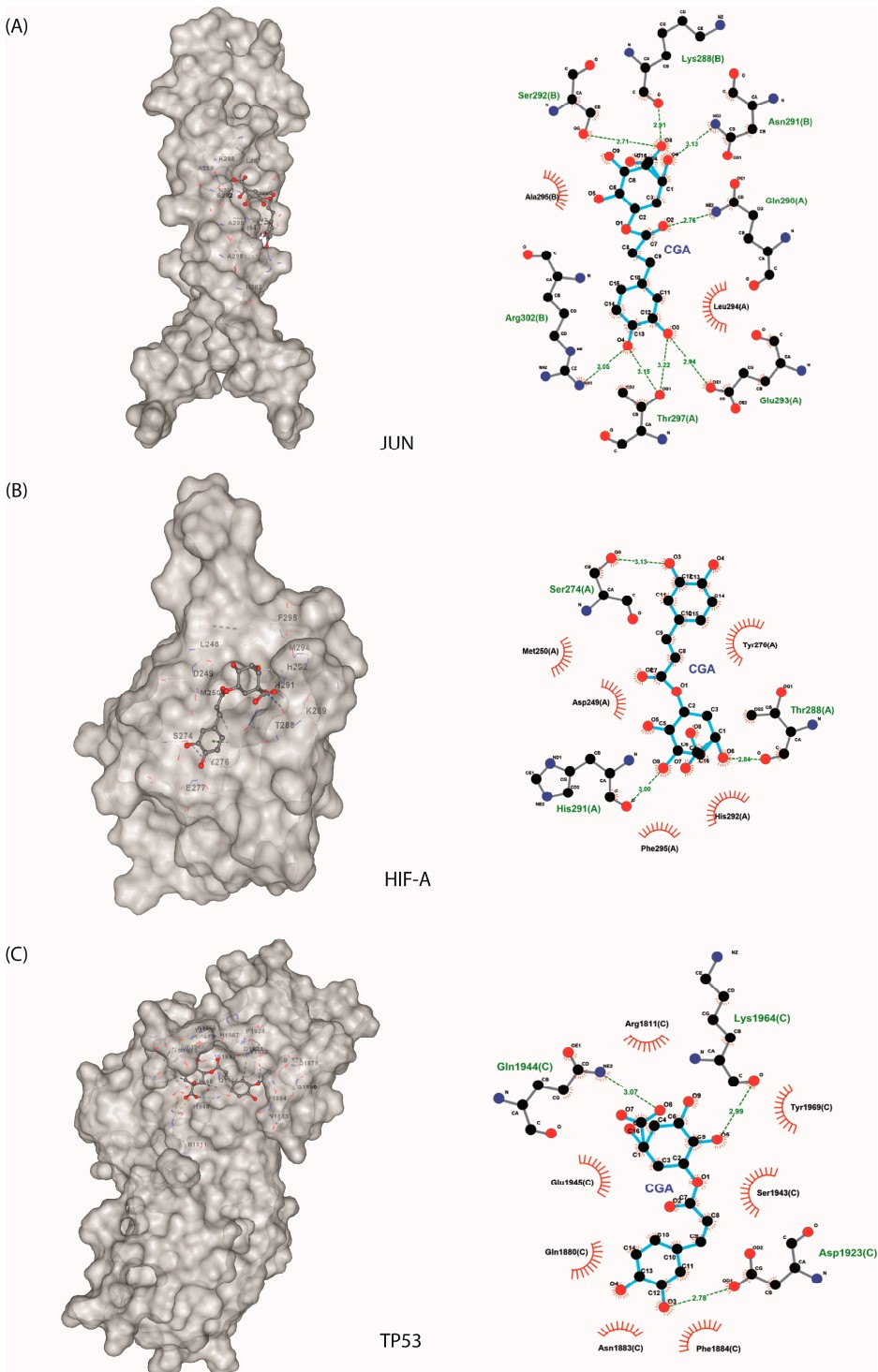

**Figure 7.** Schematic representation of the molecular docking of transcription factors. (**A**): JUN and GCA docking energy is −5.6 kcal/mol; (**B**): HIF-A and GCA docking energy is −5.8 kcal/mol; (**C**): TP53 and GCA docking energy is −7.1 kcal/mol.

In addition, in relation to the target proteins associated with vasculogenesis, the results showed that SRC with GCA had a score of −8.0 kcal/mol, with hydrogen bridges at the following amino acids: Asp404, Lys295, Met341, and Thr338. IKBKB with GCA had a score of −8.6 kcal/mol, with hydrogen bridges at the following amino acids: Cys99, Glu149, Asn150, Lys44, Asp166, and Glu97. VEGF-A with GCA had a score of −5.8 kcal/mol, with hydrogen bridges at the following amino acids: Cys57, Asn62, Asp63, and Leu66 (Figure 8).

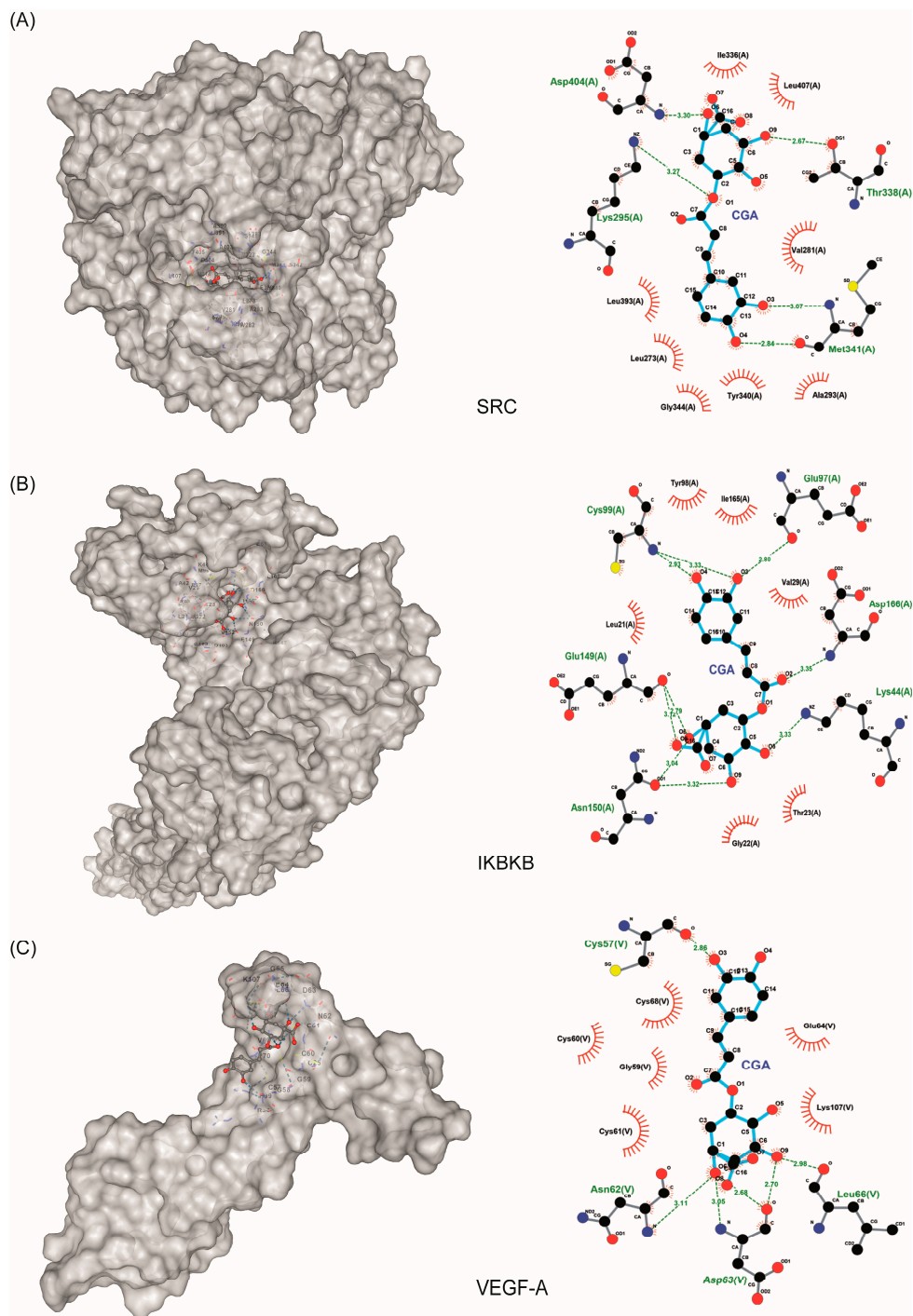

**Figure 8.** Schematic representation of molecular docking of proteins involved in vasculogenesis. (**A**): SRC and GCA docking energy is −8.0 kcal/mol; (**B**): IKBKB and GCA docking energy is −8.6 kcal/mol; (**C**): VEGF-A and GCA docking energy is −5.8 kcal/mol.

Finally, in relation to target proteins associated with apoptosis, the results showed that CASP1 with GCA had a score of −7.2 kcal/mol, with hydrogen bridges at the following amino acids: Gly242, Leu258, Arg240, Arg286, Glu390, Ile239, and Cys285. CASP3 with GCA had a score of −7.3 kcal/mol, with hydrogen bridges at the following amino acids: Cys163, Gln161, Arg64, Ser120, Arg207, Ser209, and Phe250. CASP8 with GCA had a score of −7.3 kcal/mol, with hydrogen bridges at the following amino acids: Asn407, Gly321, and Gln361 (Figure 9).

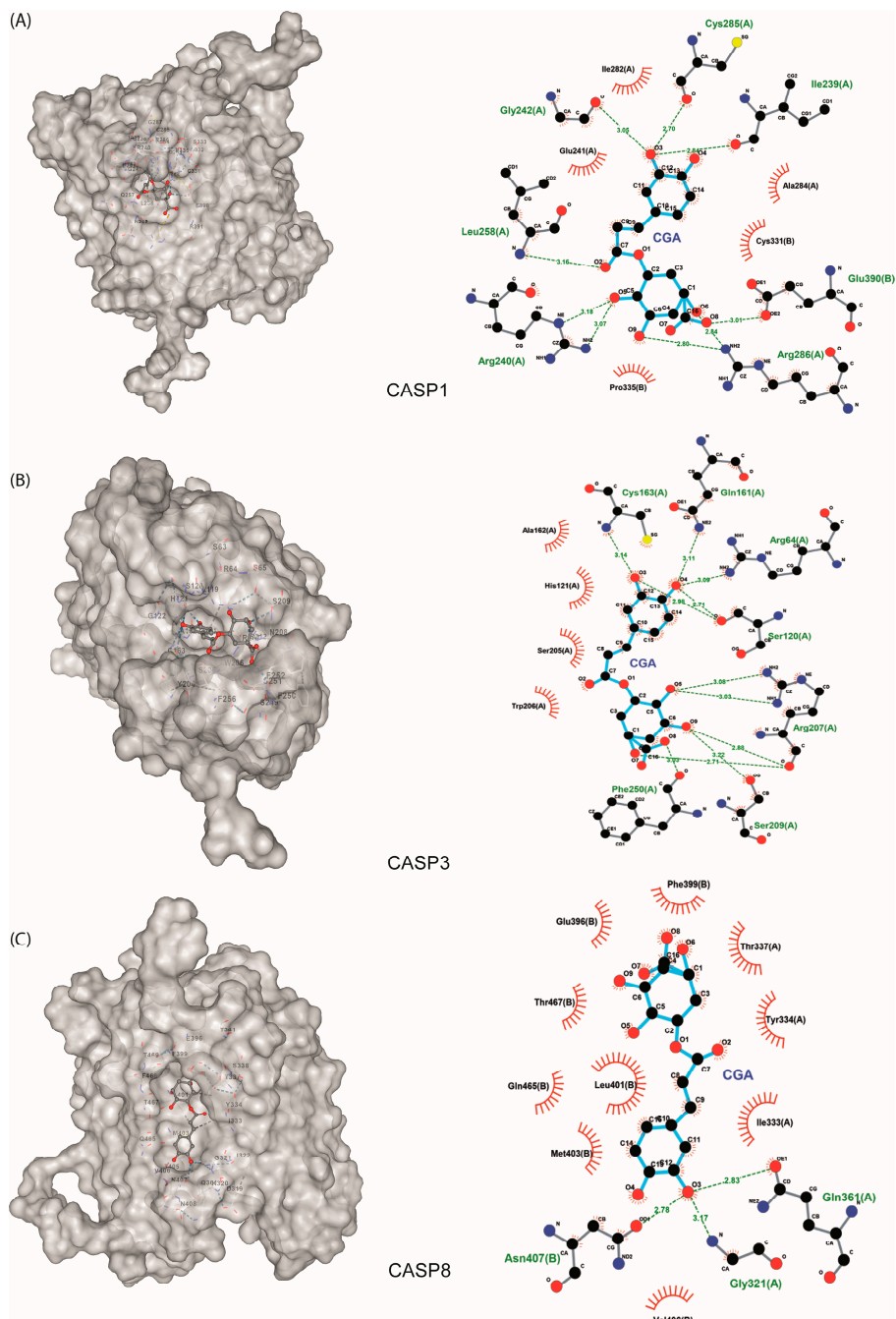

**Figure 9.** Schematic representation of molecular docking of proteins involved in apoptosis. (**A**): CASP1 and GCA docking energy is −7.2 kcal/mol; (**B**): CASP3 and GCA docking energy is −7.3 kcal/mol; (**C**): CASP8 and GCA docking energy is −7.3 kcal/mol.

## 4. Discussion

In this study, we established a network pharmacology and molecular docking approach to systematically analyze the potential targeting of CGA for PAH. From our database search, we observed 133 common targets between CGA and PAH. In addition, PPI analysis revealed 15 hub targets among the 133 common targets: TP53, HIF1A, CASP3, IL1B, JUN, MMP9, CCL2, VEGFA, SRC, IKBKB, MMP2, CASP8, NOS3, MMP1, and CASP1. The results obtained suggest that these 15 hub targets are associated mainly with pathways such as lipid and atherosclerosis, fluid shear stress and atherosclerosis, the IL-17 signaling pathway, bladder cancer, the AGE-RAGE signaling pathway in diabetic complications, Kaposi sarcoma-associated herpesvirus infection, the TNF signaling pathway, pathways

in cancer, human cytomegalovirus infection, and relaxin signaling pathway. Interestingly, our results suggest that there is an association between the main targets of CGA and lipid pathways and atherosclerosis, even though these main targets are not directly related to lipid metabolism or fat accumulation in the arteries. The association of these targets with lipid pathways and atherosclerosis could be due to the interconnection of biological processes and signaling pathways in the context of PAH. PAH is a complex disease with multiple deregulations in the pulmonary vascular system, leading to a variety of cellular and molecular responses [41]. For example, in the context of atherosclerosis, extensive research has been carried out on the role of hypoxia and its relationship with the induction of other cellular processes, such as the increase in oxidative stress, the release of proinflammatory mediators, the activation of metalloproteinases (MMPs), lipid peroxidation, and the promotion of angiogenesis [41,42]. In vitro experiments have provided evidence that the increase in HIF1$\alpha$ activity, resulting from exposure to hypoxia, can influence the modulation of lipid metabolism in vascular smooth muscle cells (VSMCs) and macrophages. This phenomenon results in a decrease in the formation of foam cells in the human monocytic cell line (U937) when stimulated with oxidized low-density lipoprotein (ox-LDL), which also correlates with the reduction in the production of proinflammatory mediators, such as interleukin-1 beta (IL-1$\beta$) [42]. In the context of PAH, it has been postulated that hypoxia triggers a series of processes related to vascular remodeling in the pulmonary arteries. These processes include a notable increase in the proliferation and migration of pulmonary artery smooth muscle cells (PASMC), the modulation of apoptosis, the formation of myofibroblasts, and the increase in the accumulation of extracellular matrix proteins. These changes are regulated by various inflammatory mediators, including endothelin 1 (ET-1), angiotensin-1 receptor, vascular endothelial growth factor (VEGF), platelet-derived growth factor (PDGF), and endothelial nitric oxide synthase (eNOS) [41,43]. In vitro experiments have provided evidence that the deletion of the St2 or HIF1$\alpha$ genes leads to a statistically significant decrease in IL-33-induced expression of HIF1$\alpha$/VEGFA/VEGFR-2 in human pulmonary artery endothelial cells (HPAEC). This phenomenon results in the inhibition of the processes associated with IL-33-induced angiogenesis in HPAECs [44]. Therefore, although the main CGA targets identified in this study may not be directly related to lipid metabolism, they may be involved in signaling cascades or molecular interactions that indirectly affect this signaling pathway.

Our study used molecular docking analysis to study and evaluate the binding affinity between CGA and 15 potential hub targets for PAH, including TP53, HIF1A, CASP3, IL1B, JUN, MMP9, CCL2, VEGFA, SRC, IKBKB, MMP2, CASP8, NOS3, MMP1, and CASP1. Encouragingly, our results indicated that CGA exhibits robust binding affinity to these core targets. The lowest binding energy was observed in the interaction between CGA and MMP2, with a value of −9.4 kcal/mol. It was followed by NOS3, IKBKB, SRC, MMP9, MMP1, CASP8, CASP3, CASP1, and TP53, with binding energies of −8.8, −8.6, −8, −7.8, −7.3, −7.3, −7.3, −7.2, and −7.1 kcal/mol, respectively. In contrast, JUN showed the highest binding energy, which was −5.6 kcal/mol, followed by HIF1A, VEGFA, CCL2, and IL1B, with binding energies of −5.8, −5.8, −5.9, and −6.6 kcal/mol, respectively. On average, the calculated binding energy was −7.23 kcal/mol, suggesting that these targets might play a crucial role in PAH. It is relevant to note that the lower the binding energy is, the higher is the estimated affinity between a ligand or drug and a protein [45]. In this context, the molecular docking results improve the accuracy of our assessment of the therapeutic potential of CGA in the treatment of PAH.

Currently, approximately 14 drugs in use have been approved by the FDA for the treatment of PAH. These drugs target the regulation of pathways associated with endothelin, prostacyclin, and nitric oxide (NO), which play a key role in the pathophysiological mechanism of PAH. Our findings suggest that CGA may regulate some of these pathways, raising the possibility that it may have synergistic effects with certain FDA-approved drugs [46,47]. For example, sildenafil, a phosphodiesterase-5 (PDE-5) inhibitor, gained FDA approval in 2005 for the treatment of PAH. The PDE-5 enzyme is found in the smooth

muscle cells of the pulmonary arteries and plays a role in the breakdown of NO, which can lead to the constriction of the pulmonary arteries and ultimately PAH. Sildenafil acts by inhibiting PDE5, which allows an increased accumulation of NO in the pulmonary arteries. This leads to the relaxation of the smooth muscles of the arteries, reducing pulmonary blood pressure and improving blood flow in the lungs [47]. Recent research has suggested that hypoxia is related to decreased NO expression and function. HIF-1$\alpha$ has been shown to play an essential role in the development of the hypoxic state in PAH [48,49]. In this context, our results indicate that CGA might inhibit HIF-1$\alpha$ activity, which could result in a reduction in the hypoxic state. This reduction could increase NO levels, which could have a beneficial effect on reducing hypoxia-associated pulmonary vascular remodeling in PAH. Interestingly, our study suggests that CGA may act on key molecules associated with the inflammatory process in PAH, such as IL1$\beta$ and CCL2. For example, IL1$\beta$ is a proinflammatory cytokine that belongs to the IL-1 family of interleukins consisting of 11 members involved in inflammatory processes [50]. The interaction between IL-1$\beta$ and its receptor IL-1R1, together with the accessory protein IL-1RAcP, has been shown to play an important role in promoting inflammatory responses. This interaction favors the positive activation of the expression of several genes associated with events related to subsequent inflammatory processes, as well as disease progression and tissue injury [50,51]. IL-1$\beta$ has been shown to play an essential role in PAH, and recent evidence has suggested that serum IL1$\beta$ levels are significantly elevated in PAH patients and correlate with a worse outcome [50,52]. In addition, studies in a mouse model of PH demonstrated that the inhibition of IL-1$\beta$ with the anti-inflammatory IL-1Ra prevents the increase in pulmonary vascular resistance and protects against changes in vascular morphology [53]. Similarly, treatment with an IL-1Ra antagonist inhibited the development of chronic pulmonary hypertension in a rat model of PH [54]. Moreover, human studies have shown that suppression of IL-1R1 with the IL1$\beta$ receptor antagonist anakinra alleviated PH in a patient with adult Still's disease and had a positive effect on reducing inflammation in PAH patients [55,56]. Together, our results suggest that the possible binding of CGA-IL1$\beta$ could play an essential role in the inflammatory process associated with the development of various forms of PH and may be considered a promising therapeutic molecule for PAH.

CCL2, also known as monocyte chemoattractant protein-1 (MCP-1), is a protein with chemoattractant properties for different cell types in the area of inflammation to eliminate pathogens and repair tissues. Although its main origin is thought to be immune system cells, recent studies have revealed that smooth muscle cells, endothelial cells, and fibroblasts can also synthesize and release CCL2. This wide cellular distribution contributes to the diversity of CCL2 functions in various physiological and pathological processes [57,58]. CCL2 and its main chemokine receptor CCR2 are involved in the chronic inflammation of various diseases, such as some types of cancer, rheumatic diseases, and lung diseases; therefore, it has been suggested that it participates in cellular processes such as proliferation, survival, differentiation, migration, invasion, and angiogenesis [57–61]. The available evidence suggests that CCL2 is upregulated in the lung tissue and plasma of PAH patients compared to their respective controls [62]. It is striking that in the context of PAH, a significant increase in CCL2 expression has been detected in pulmonary vascular endothelial cells. CCL2 has the ability to attract inflammatory cells present in the circulation and acts as a stimulatory factor for pulmonary arterial smooth muscle cells (PASMCs). Furthermore, both PASMCs and perivascular macrophages in PAH patients exhibited elevated levels of CCR2 receptors compared to control subjects. Indeed, the presence of CCR2 has been found to be critical in initiating and enhancing PASMC proliferation. Therefore, our results suggest that the regulation of CCL2 through CGA could play an essential role in the inhibition of the inflammatory process involved in the development of PAH and that CGA could be a potential molecule with a therapeutic effect [62,63].

In our molecular docking analysis, we showed that CGA can modulate proteins such as HIF-1$\alpha$, VEGF, SRC, eNOS, and IKK. Hypoxia refers to a decrease in the availability of oxygen, whether it occurs in an entire organism or an individual cell. Previous studies

have indicated that hypoxia can reduce the degradation of HIF-1$\alpha$, a protein that plays a crucial role in gene regulation. HIF-1$\alpha$ is responsible for promoting the expression of various genes, including VEGF and eNOS, which are essential for cell proliferation [64–66]. Additionally, SRC is necessary for the expression of HIF-1$\alpha$ in vascular smooth muscle cells (VSMCs) [67]. HIF-1 is a key transcription factor mediating survival during hypoxic conditions and consists of inducible HIF-1$\alpha$ and constitutive HIF-1$\beta$, where HIF-1$\alpha$ is stabilized and translocated to the nucleus and binds to the hypoxia-responsive element (HRE) to induce the transcription of many genes involved in cell survival that promote vascular development, glycolytic metabolism, and cell cycle control [64,68,69]. HIF-1$\alpha$ has been shown to play an essential role in PAH since it has been observed to be overexpressed in the lung tissues of patients with PAH and also in other pulmonary diseases, such as lung cancer, chronic obstructive pulmonary disease (COPD), pulmonary fibrosis, and pulmonary hypertension [48,70].

VEGF is an endothelial cell-specific mitogen and a potent angiogenic peptide that is secreted by a variety of cell types and is involved in vascular remodeling in PAH [71]. In addition, a correlation has been found between elevated VEGF levels and systolic pulmonary artery pressure in the serum of patients with PAH in systemic sclerosis, thus suggesting an important role in the pathogenesis [72]. Additionally, in a model of alpha-naphthylthiourea (ANTU)-induced PAH, which is an animal model that mimics most of the clinicopathological characteristics of human disease, Linlin Wang et al. showed that VEGF modulates eNOS activity to inhibit endothelial cell apoptosis [73].

eNOS synthesizes NO in endothelial cells. eNOS plays an important role in angiogenesis, participating as a component of the VEGF pathway and regulating the structural formation of the endothelium [74]. eNOS is an important signaling molecule in the cardiopulmonary vascular system and is an important mediator of alveolarization and lung growth. One of the major inflammatory mediators involved in inflammation and angiogenesis is NO, which is a potent vasodilator and mediator of oxidative stress and is a free radical synthesized in cells by the oxidation of L-arginine by eNOS [75]. In addition, decreased NO production contributes to the pathogenesis of PAH in different cell lines and animal models, such as the monocrotaline-induced PAH model. eNOS is underexpressed, and therapeutic treatments increase its expression and NO production, protecting against the injury induced by PAH [76–78].

Src, known as c-Src in humans, is a tyrosine kinase associated with the cell or endosomal membranes and is involved in many cell functions, such as proliferation, differentiation, motility, and adhesion; it is implicated in several diseases as well as those involving the lung, such as lung cancer, pulmonary fibrosis, and PAH [79–81]. Interestingly, Src is essential for the activation of HIF-1$\alpha$ from reactive oxygen species (ROS) generated in the mitochondria of VSMCs [67]. Finally, Qun-Yi Li et al. showed that CGA attenuated the hypoxia-induced effect on PASMC proliferation via the c-Src/Shc/Grb2/ERK2 signaling pathway. In addition, CGA inhibited the monocrotaline-induced PAH model in rats [22]. Our results showed a possible binding between CGA-c-Src, having the fourth position in the docking values, and our results reinforce the study of Qun-Yi Li et al. and may indicate that there is a direct binding of CGA-c-Src and not a mere upstream inhibition of c-Src.

The inhibitor of nuclear factor kappa B kinase (IKK) is encoded by the IKBKB gene and is a complex comprising three subunits: IKK$\alpha$, IKK-$\beta$, and IKK-$\gamma$. IKK-$\beta$ has been reported as a key kinase in NF-$\kappa$B signaling. NF-$\kappa$B has crucial roles in the regulation of immune and inflammatory responses, as well as in cell growth and EMT processes. The rapid activation of NF-$\kappa$B in response to various stimuli enables the transcription of genes encoding cytokines, chemokines, and membrane proteins [82–84]. Due to chronic inflammation, NF-$\kappa$B signaling is activated in endothelial cells, smooth muscle cells, macrophages, and lymphocytes of PAH patients. In this context, IKK-$\beta$ is phosphorylated and degraded, and this activation of NF-$\kappa$B results in the release of IL6, IL8, and MCP-1 [52,85–87]. Our results showed a possible binding between CGA-IKBKB, with the third position in the docking values. Based on this binding energy we could hypothesize that the binding between

CGA-IKBKB could inhibit the phosphorylation of IKBKB and therefore the degradation, resulting in the inhibition of NF-κB and the release of IL6, IL8, and MCP-1.

Another characteristic of PAH patients is the increase in vascular stiffness due to ECM deposition and remodeling in the vascular system, where inflammation and ROS play an important role in triggering these characteristics. Several ECM proteins are increased in the pulmonary vasculature, such as collagens, elastins, tenascin, and fibronectin [4,88]. Physiologically, the regulation of ECM remodeling is mediated by a series of proteolytic enzymes, such as metalloproteases, serine elastases, lysyl oxidases (LOXs), tissue inhibitors of metalloproteinases (TIMPs), and MMPs. MMPs are zinc-dependent endopeptidases that have been found to be overexpressed in the pulmonary arteries of different models and in PAH patients, and it has been suggested that their inhibition could serve as a therapeutic strategy for PAH [4,89,90]. In our docking analysis, we found that CGA can bind to MMPs-1, -2, and -9. MMP-1, -2, and -9 seem to be overexpressed in lung tissue, plasma, and urine of patients and animal models of PAH [89,91–93]. In addition, in the monocrotaline model, all-trans retinoic acid was shown to reduce the overexpression of MMP-1 mRNA, which reduced the hydroxyproline content and the mean pulmonary artery pressure [92]. In 2019, Peiyuan Bai et al. showed that MMP-2 is activated by legumain and that MMP-2 hydrolyzes and activates the precursor TGF-β1 in hypoxia plus SU5416 and monocrotaline animal models [94], and TGF-β1 seems to be a crucial molecule in the pathogenesis of pulmonary vascular remodeling [95]. In children with PAH, it has been suggested that there is an association between ECM remodeling and MMP-9 with cardiac hemodynamics and proximal pulmonary arterial stiffness [90]. In the monocrotaline-induced HPA model, a transgenic mouse with MMP-9 overexpression exhibited pulmonary vascular remodeling and pulmonary hypertension [91]. Therefore, the available data suggest that MMPs play an essential role in the pathophysiological process of PAH and may be considered a promising therapeutic target for the disease. Our results show that CGA may be considered a promising therapeutic molecule, since CGA binding to MMP-1, 2 and 9 could inhibit them, resulting in the inhibition of ECM remodeling.

On the other hand, there are 18 caspases that are divided into inflammatory and apoptotic caspases. Inflammatory caspases comprise 1, 4, 5, 11, 12, and 13. The initiator caspases are 2, 8, 9, and 10, and the effector caspases are 3, 6, and 7 [96]. Caspase-1 is the mediator of inflammation through the activation of cytosines such as IL-18 and Il-1 β. In hypoxia, this activation has been investigated in caspase-1-deficient mice, and they showed lower pressure and low muscularization in pulmonary arteries. In addition, it was shown that the addition of IL-18 or IL-1β to caspase-1-deficient pulmonary arteries retained smooth muscle cell proliferation. Thus, caspase-1 inhibition is proposed as a potential target for the treatment of PAH [97]. In apoptosis, two pathways are known, extrinsic and intrinsic, which result in the activation of different caspases [98]. Therefore, the cell death receptors are apical signaling pathways in the extrinsic pathway, which are caspases 8 and 10 [99]. The role of caspase 8 in hypoxia-induced PAH has been studied, and an increase in inactive and active caspase 8 has been observed in lung tissue, which was corroborated by pharmacological blockade and gene knockout; furthermore, increased caspase 8 expression elevated IL-1β activity in macrophages, and its blockade could alleviate PAH by inhibiting NLRP3/IL-1β signaling [100]. The activation of caspase 8 and 9 has been demonstrated in pulmonary artery smooth muscle cells as a result of bone morphogenetic protein (BMP) interaction [101].

As a member of the family of effector caspases, the role of caspase 3 has been analyzed by several research groups in PAH, such as in the model of PAH induced by monocrotaline. When treated with formononetin (FMN), a natural isoflavone, a protective effect against PAH was observed by exerting resistance to apoptosis [102]. Furthermore, endothelin-1 treatment enhanced paclitaxel-mediated caspase-3 expression in a model of cultured neonatal rat PASMCs [103]. A described apoptosis axis revealed that using apoptotic stimuli, human pulmonary arterial endothelial cells (PAECs) showed a strong induction of

the caspase-3-related programmed cell death 4 (PDCD4) pathway, which was reversible by direct silencing of PDCD4 [104].

JUN is a major component of the transcription factor AP-1, a heterodimer formed by members of the Jun and Fos protein families. AP-1 plays an essential role in diverse biological processes, such as embryonic development, differentiation, survival, apoptosis, proliferation, and cell migration. Recent research has studied the relationship of AP-1 to the development of chronic diseases such as cancer, and these investigations have focused on AP-1 as a therapeutic target because of its important role as a critical regulator of proinflammatory genes related to tissue remodeling [105–107]. Recent studies have demonstrated that enalapril treatment suppressed the development of PH induced by bleomycin administration in a murine model, and enalapril treatment inhibited NF-κB and AP-1 activation and decreased the expression of collagen and TNF-α [108]. Moreover, high extracellular mobility group box 1 (HMGB1) was shown to maintain overexpression in patients with PH compared to the respective controls, and in vitro experiments suggested that HMGB1 enhanced the proliferation of PASMCs and PAECs by promoting the activation of the downstream AP-1 complex proteins c-Fos and c-Jun, whereas c-Jun silencing suppressed HMGB1-induced proliferation in PASMCs [109].

The p53 protein, encoded by the TP53 gene, acts as an important tumor suppressor by inhibiting the growth of multiple tumors. p53 functions as a specific transcription factor that regulates the expression of target genes that play an essential role in biological functions such as promoting cell cycle arrest, apoptosis, and DNA repair [110]. Therefore, the central role of p53 has been extensively studied in various types of cancer, even though the relationship of p53 with other chronic diseases has now been studied [110,111]. Recent studies have evaluated the possible role of p53 in the development of PAH [112]. For example, a study performed in a murine model of hypoxia-induced PH showed that p53 knockout mice developed more severe PH in response to chronic hypoxia than wild-type mice [112]. On the other hand, the use of Nutlin-3a had a partial effect in reversing chronic hypoxia-induced PH-associated lung damage in mice. Nutlin-3a treatment resulted in a significant increase in p21-labeled PASMC senescent cells, as well as in the levels of p53, p21, and MDM2 proteins in lung tissue. The results obtained suggest that to prevent PH damage, it was necessary to stabilize p53 in the lungs and increase p21 expression, as evidenced by the lack of effects of Nutlin-3a in mice exposed to hypoxia that lacked p53 $^{-/-}$ and p21 $^{-/-}$ [113]. Furthermore, it was shown that the inactivation of p53 by treatment with pifitrin-α (PFT, an inhibitor of p53 activity) was sufficient to induce pulmonary vascular remodeling associated with PH and to aggravate MCT-induced PH in rats. Together, these results suggest that the p53 pathway may be involved in the pathogenesis of PH [114]. The available evidence suggests that treatment with CGA present in coffee husk extract helped to reduce the levels of oxidative stress in human mesenchymal stem cells (hMSCs) by increasing the mRNA expression of antioxidants CYP1A, GSH, GSK-3β, and GPX and tumor suppressor genes such as Cdkn2A and p53 [115].

In summary, our study presents promising findings and suggests that CGA may act on key targets and pathways associated with PAH, such as inflammation, vascular remodeling, and hypoxia-related processes. These findings support the potential of CGA as a therapeutic agent for PAH by modulating these molecular targets. However, it is important to highlight some limitations inherent to this investigation. First, the association of some of the key targets with pathways such as lipid- and atherosclerosis-related pathways, which are not directly related to the pathophysiology of PAH, was observed. These observations underscore the need for further investigations to fully understand the scope of CGA action in PAH and to validate its potential efficacy in preclinical and clinical models. Furthermore, the study is based on in silico analysis and bioinformatics data, which requires experimental confirmation and further validation in in vitro and in vivo studies. Despite these limitations, the results of this research lay a solid foundation for future research in the field of PAH and new drug development.

## 5. Conclusions

In conclusion, our study provides preliminary evidence on the underlying molecular mechanism of CGA in the treatment of PAH. The findings suggest that CGA could be a promising option for the development of new drugs for PAH. However, further in vitro and in vivo studies are required to validate and optimize these findings and to fully understand the role of CGA in PAH.

**Supplementary Materials:** The following supporting information can be downloaded at: https://www.mdpi.com/article/10.3390/jvd3010002/s1, Table S1. Potential targets for CGA from the SwissTargetPrediction database. Table S2. Potential targets for CGA from the GeneCard database. Table S3. Potential targets for PAH from the GeneCard database. Table S4. Potential targets for PAH from the DisGeNET database. Table S5. Overlapping CGA and PAH targets: 133 common targets. Table S6. Gene ontology enrichment and KEGG pathway analysis of the 15 hub targets between CGA and PAH.

**Author Contributions:** Conceptualization, J.C.S.-Á., J.M.V.-E. and R.B.-H.; methodology, J.C.S.-Á., J.M.V.-E. and R.B.-H.; validation, J.C.S.-Á. and J.M.V.-E.; formal analysis, J.C.S.-Á. and J.M.V.-E.; investigation, J.C.S.-Á., J.M.V.-E. and R.B.-H.; resources, J.C.S.-Á., J.M.V.-E. and R.B.-H.; data curation, J.C.S.-Á. and J.M.V.-E.; writing—original draft preparation, J.C.S.-Á., J.M.V.-E. and R.B.-H.; writing—review and editing, J.C.S.-Á., J.M.V.-E. and R.B.-H.; visualization, J.C.S.-Á. and J.M.V.-E.; supervision, J.C.S.-Á., J.M.V.-E. and R.B.-H.; project administration, J.C.S.-Á., J.M.V.-E. and R.B.-H.; funding acquisition, J.C.S.-Á., J.M.V.-E. and R.B.-H. All authors have read and agreed to the published version of the manuscript.

**Funding:** This research was funded by the Consejo Nacional de Ciencia y Tecnología CONACyT, supported by the strengthening and development of Scientific and Technological Infrastructure 2016 grant (No. 270189) to R.B.H and by CONACyT National Scholarships (Traditional) 2020-2 grant (No. 772855) to J.M.V.-E. and a grant (No.773019) to J.C.S.-Á.

**Institutional Review Board Statement:** Not applicable.

**Informed Consent Statement:** Not applicable.

**Data Availability Statement:** All data generated or analyzed during this study are included in this published article and its supplementary information files.

**Conflicts of Interest:** The authors declare that they have no conflicts of interest.

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
