# Peer review of "Evaluation of the Molecular Mechanism of Chlorogenic Acid in the Treatment of Pulmonary Arterial Hypertension Based on Analysis Network Pharmacology and Molecular Docking"

_2813-2475, doi:10.3390/jvd3010002_

Round 1

Reviewer 1 Report

Comments and Suggestions for Authors

Evaluation of the molecular mechanism of chlorogenic acid in the treatment of pulmonary arterial hypertension based on analysis network pharmacology and molecular docking.

Minor issues:

·         On the front page, line 26, change ACG to CGA.

·         Page 3, line 101: What is IPF? Define MCC on line 107 as on page 5, line 159, then use the abbreviation.

·         In the 2.5 section, change GO for gene ontology and proceed with abbreviations.

·         Page 6, figure legend: change CMM to MCC.

·         TP53 has been used throughout the paper, and then on page 9, line 250, page 10 figure and figure legend line 256, there is TP53BP1. Is it the same molecule?

 The work is exciting; its results support the pathways known to be altered in PAH. It is consistent with the possible routes to explore with CGA. It is a significant study.

How can it be explained that the results from the top 15 targets suggest that the lipid and atherosclerosis pathways are associated? All other suggestions are logical based on the damage present in the disease that needs to be targeted.

None of the targets are directly related to lipid metabolism or fat accumulation in the arteries.

Finally, depending on their affinity, it would be possible to determine the concentration and treatment time.

Author Response

We sincerely thank the reviewer for your careful review and valuable comments. His observations and suggestions have been of great help in improving the quality and presentation of our manuscript. We have taken into account each of the points he mentioned.

Minor issues:

  1. On the front page, line 26, change ACG to CGA.

Answer: We thank the reviewer for your detailed comments and suggestions. We have taken your suggestion into consideration and have made the requested correction. On the first page, line 28, we have changed "ACG" to "CGA" to ensure accuracy and consistency in the presentation of our data.

  1. Page 3, line 101: What is IPF? Define MCC on line 107 as on page 5, line 159, then use the abbreviation.

Answer: We appreciate your constructive comments and feedback on our manuscript. We have addressed your concerns as follows:

1.- Regarding page 3, line 101, we acknowledge that we made an error in mentioning "IPF" instead of "PAH". We appreciate your pointing out this error and have made the appropriate correction. We have replaced "IPF" with "PAH" on line 101 to accurately reflect the context of our study.

2.- In addition, on line 107 on page 3, we have defined "MCC" as "Maximum Clique Centrality" as requested, and have continued to use the abbreviation "MCC" for consistency throughout the manuscript.

  1. In the 2.5 section, change GO for gene ontology and proceed with abbreviations.

Answer: We thank the reviewer for your comments and suggestions. We have addressed your suggestion and made the requested correction in section 2.5. We have changed "GO" to "Gene Ontology" and will continue to use the abbreviation "GO" from that point forward for conciseness and consistency.

  1. Page 6, figure legend: change CMM to MCC.

Answer: We thank the reviewer for your detailed comments and suggestions. We have addressed your suggested correction in the figure legend of figure 3 on page 6.

We have changed "CMM" to "MCC" in the figure legend to accurately reflect the correct abbreviation to ensure clarity and consistency in our manuscript.

  1. TP53 has been used throughout the paper, and then on page 9, line 250, page 10 figure and figure legend line 256, there is TP53BP1. Is it the same molecule?

Answer: We appreciate your review and valuable suggestions for improving our manuscript.  We have noted your observation and appreciate your pointing out the discrepancy in the use of the abbreviations "TP53" and "TP53BP1".

We regret any confusion this discrepancy may have caused and assure that the proper abbreviation is TP53 which was one of the central CGA targets in our study. Therefore, molecular docking analysis was performed with TP53 and its crystal structure obtained from PDB with the ID:1KZY. We confirm that this abbreviation was homogenized in the manuscript to ensure consistency and accuracy in its presentation.

  1. The work is exciting; its results support the pathways known to be altered in PAH. It is consistent with the possible routes to explore with CGA. It is a significant study.

How can it be explained that the results from the top 15 targets suggest that the lipid and atherosclerosis pathways are associated? All other suggestions are logical based on the damage present in the disease that needs to be targeted.

None of the targets are directly related to lipid metabolism or fat accumulation in the arteries.

Answer: We sincerely thank you for your kind words and support of our work. We are pleased that you find our study exciting and meaningful.

Regarding your question about the association of primary targets with lipid pathways and atherosclerosis, we appreciate the opportunity to clarify this. Indeed, in our results, we observed an association of the main targets identified with these pathways, even though they are not directly related to lipid metabolism or fat accumulation in the arteries. Therefore, this point has been taken into account to be addressed in the discussion section of our manuscript along the lines 320-350 as shown below:

Interestingly, our results suggest that there is an association between the main targets of CGA and lipid pathways and atherosclerosis, even though these main targets are not directly related to lipid metabolism or fat accumulation in the arteries. The association of these targets with lipid pathways and atherosclerosis could be due to the interconnection of biological processes and signaling pathways in the context of PAH. PAH is a complex disease with multiple deregulations in the pulmonary vascular system, leading to a variety of cellular and molecular responses [41]. For example, in the context of atherosclerosis, extensive research has been carried out on the role of hypoxia and its relationship with the induction of other cellular processes, such as the increase in oxi-dative stress, the release of proinflammatory mediators, the activation of metalloproteinases (MMPs), lipid peroxidation and the promotion of angiogenesis [41,42]. In vitro experiments have provided evidence that the increase in HIF1α activity, resulting from exposure to hypoxia, can influence the modulation of lipid metabolism in vascular smooth muscle cells (VSMCs) and macrophages. This phenomenon results in a decrease in the formation of foam cells in the human monocytic cell line (U937) when stimulated with oxidized low-density lipoprotein (ox-LDL), which also correlates with the reduction in the production of proinflammatory mediators, such as interleukin-1 beta (IL-1β) [42]. In the context of PAH, it has been postulated that hypoxia triggers a series of processes related to vascular remodelling in the pulmonary arteries. These processes include a notable increase in the proliferation and migration of pulmonary artery smooth muscle cells (PASMC), the modulation of apoptosis, the formation of myofibroblasts and the increase in the accumulation of extracellular matrix proteins. These changes are regulated by various inflammatory mediators, including endothelin 1 (ET-1), angiotensin-1 receptor, vascular endothelial growth factor (VEGF), plate-let-derived growth factor (PDGF), and endothelial nitric oxide synthase (eNOS) [41,43]. In vitro experiments have provided evidence that deletion of the St2 or HIF1α genes leads to a statistically significant decrease in IL-33-induced expression of HIF1α/VEGFA/VEGFR-2 in human pulmonary artery endothelial cells (HPAEC). This phenomenon results in the inhibition of the processes associated with IL-33-induced angiogenesis in HPAECs [44]. Therefore, although the main CGA targets identified in this study may not be directly related to lipid metabolism, they may be involved in signaling cascades or molecular interactions that indirectly affect this signaling pathway.

  1. Finally, depending on their affinity, it would be possible to determine the concentration and treatment time.

Answer: We appreciate the reviewer's very important comment about our manuscript. Your comments are very helpful in improving the quality of our work.

In response to your comment about determining concentration and treatment time as a function of affinity, we would like to emphasize that our research focuses specifically on assessing the affinity between drugs and their target proteins using molecular docking and network analysis techniques. Importantly, affinity refers to the strength of the interaction between the drug and the protein. The higher the affinity, the stronger the binding, which in turn can influence concentration and treatment time.

Pinzi L, Rastelli G. Molecular Docking: Shifting Paradigms in Drug Discovery. Int J Mol Sci. 2019 Sep 4;20(18):4331. doi: 10.3390/ijms20184331. PMID: 31487867; PMCID: PMC6769923. (Reference provided for peer review only)

Terefe, E.M.; Ghosh, A. Molecular Docking, Validation, Dynamics Simulations, and Pharmacokinetic Prediction of Phytochemicals Isolated From Croton dichogamus Against the HIV-1 Reverse Transcriptase. Bioinformatics and biology insights 2022, 16, 11779322221125605, doi:10.1177/11779322221125605. (Reference provided for peer review only)

However, there are other key points to consider in determining concentration and treatment time, such as pharmacokinetics and pharmacodynamics. Although we recognize the importance of pharmacokinetics and pharmacodynamics in clinical practice, unfortunately we do not have data related to these aspects in the present study.

We understand that pharmacokinetics and pharmacodynamics are fundamental to the optimization of drug therapy, and while we cannot provide specific data in this area in this article, our findings regarding the affinity between drugs and their protein targets may serve as a valuable basis for further research focusing on these aspects.

Reviewer 2 Report

Comments and Suggestions for Authors

Thank you for the opportunity to review this paper.

This study evaluate the potential of CGA for the treatment of PAH, using two methods of investigation, obtaining 133 common to CGA and PAH targets (2.3% of the total) and 15 hub targets. The futher analysis of the hub targets of CGA and PAH using GO terms, the KEGG pathway and the molecular docking reveals  CGA targets for PAH

This paper has an extended discussion, detailed, about the role of all these factors in the pathophysiology of PAH, but there is a limited discussion of the results of this study. Please add more comments about the rasults of the study. For example, the results about the MMPs ,(especially MMP2), with better binding energy, are presented in the table1, figures 5,6, but  the interpretation of these results are limited (lines 396-401), with emphasis on the bibliography data. please comment further the results, the possible importance  of these findings as common targets with the targets of the current specific therapy agents for PAH (with the same pathways). Please relate the results about the 15 hub targets (i.e. eNOS) in this study with the targets of PAH therapy (ERA, NO-pathway, PG pathway) and if there is a possible minimal synergistic effect 

  Please add to the discussion the limitation of these study

Author Response

We sincerely thank the reviewer for your careful review and valuable comments. His observations and suggestions have been of great help in improving the quality and presentation of our manuscript. We have taken into account each of the points he mentioned.

Thank you for the opportunity to review this paper.

This study evaluate the potential of CGA for the treatment of PAH, using two methods of investigation, obtaining 133 common to CGA and PAH targets (2.3% of the total) and 15 hub targets. The futher analysis of the hub targets of CGA and PAH using GO terms, the KEGG pathway and the molecular docking reveals  CGA targets for PAH

  1. This paper has an extended discussion, detailed, about the role of all these factors in the pathophysiology of PAH, but there is a limited discussion of the results of this study. Please add more comments about the rasults of the study. For example, the results about the MMPs ,(especially MMP2), with better binding energy, are presented in the table1, figures 5,6, but the interpretation of these results are limited (lines 396-401), with emphasis on the bibliography data. please comment further the results, the possible importance  of these findings as common targets with the targets of the current specific therapy agents for PAH (with the same pathways). Please relate the results about the 15 hub targets (i.e. eNOS) in this study with the targets of PAH therapy (ERA, NO-pathway, PG pathway) and if there is a possible minimal synergistic effect.

Answer: We welcome your comments and suggestions, and we are happy to address your concerns regarding the discussion of the results of our study. We recognize that the discussion of the results is a crucial part of the article and are committed to providing a more detailed interpretation of our findings.

In response to your suggestion, we have expanded our discussion of the results of our molecular docking study in greater depth and their potential relevance in the context of current therapeutic targets for PAH. We will also establish a clearer connection between the results on the top 15 targets identified in our study and the targets of PAH therapy, such as the nitric oxide (NO) pathway.

Therefore, these points have been taken into account to be addressed in the discussion section of our manuscript along lines 351-383 as shown below:

Our study used molecular docking analysis to study and evaluate the binding affinity between CGA and 15 potential hub targets for PAH, including TP53, HIF1A, CASP3, IL1B, JUN, MMP9, CCL2, VEGFA, SRC, IKBKB, MMP2, CASP8, NOS3, MMP1, and CASP1. Encouragingly, our results indicated that CGA exhibits robust binding affinity to these core targets. The lowest binding energy was observed in the interaction between CGA and MMP2, with a value of -9.4 kcal/mol. It was followed by NOS3, IKBKB, SRC, MMP9, MMP1, CASP8, CASP3, CASP1 and TP53, with binding energies of -8.8, -8.6, -8, -7.8, -7.3, -7.3, -7.3, -7.2 and -7.1 kcal/mol, respectively. In contrast, JUN showed the highest binding energy, which was -5.6 kcal/mol, followed by HIF1A, VEGFA, CCL2 and IL1B, with binding energies of -5.8, -5.8, -5.9 and -6.6 kcal/mol, respectively. On average, the calculated binding energy was -7.23 kcal/mol, suggesting that these targets might play a crucial role in PAH. It is relevant to note that the lower the binding energy is, the higher the estimated affinity between a ligand or drug and a protein [45]. In this context, the molecular docking results improve the accuracy of our assessment of the therapeutic potential of CGA in the treatment of PAH.

Currently, approximately 14 drugs in use have been approved by the Food and Drug Administration (FDA) for the treatment of PAH. These drugs target the regulation of pathways associated with endothelin, prostacyclin, and nitric oxide (NO), which play a key role in the pathophysiological mechanism of PAH. Our findings suggest that CGA may regulate some of these pathways, raising the possibility that it may have synergistic effects with certain FDA-approved drugs [46,47]. For example, sildenafil, a phos-phodiesterase-5 (PDE-5) inhibitor, gained FDA approval in 2005 for the treatment of PAH. The PDE-5 enzyme is found in the smooth muscle cells of the pulmonary arteries and plays a role in the breakdown of NO, which can lead to constriction of the pulmonary arteries and ultimately PAH. Sildenafil acts by inhibiting PDE5, which allows increased accumulation of NO in the pulmonary arteries. This leads to relaxation of the smooth muscles of the arteries, reducing pulmonary blood pressure and improving blood flow in the lungs [47]. Recent research has suggested that hypoxia is related to decreased NO expression and function. HIF-1α has been shown to play an essential role in the development of the hypoxic state in PAH [48,49]. In this context, our results indicate that CGA might inhibit HIF-1α activity, which could result in a reduction in the hypoxic state. This reduction could increase NO levels, which could have a beneficial effect on reducing hypoxia-associated pulmonary vascular remodelling in PAH.

  1. Please add to the discussion the limitation of these study

Answer: We welcome your review and suggestions. We recognize the importance of identifying and communicating the limitations of our study in order to provide a balanced and accurate view of the research conducted.

We have limitations that we consider pertinent to mention in the corresponding discussion section of the manuscript. Some of the limitations we have identified are included in lines 580-592 of our manuscript, as shown below:

In summary, our study presents promising findings and suggests that CGA may act on key targets and pathways associated with PAH, such as inflammation, vascular remodelling, and hypoxia-related processes. These findings support the potential of CGA as a therapeutic agent for PAH by modulating these molecular targets. However, it is important to highlight some limitations inherent to this investigation. First, the association of some of the key targets with pathways such as lipid- and atherosclerosis-related pathways, which are not directly related to the pathophysiology of PAH, was observed. These observations underscore the need for further investigations to fully understand the scope of CGA action in PAH and to validate its potential efficacy in preclinical and clinical models. Furthermore, the study is based on in silico analysis and bioinformatics data, which requires experimental confirmation and further validation in in vitro and in vivo studies. Despite these limitations, the results of this research lay a solid foundation for future research in the field of PAH and new drug development.

Reviewer 3 Report

Comments and Suggestions for Authors

1.       Authors need to establish the rationale for the use of chlorogenic acid in pulmonary arterial hypertension. Is it a random study or based on previous experimental results.

2.       Network pharmacology and Molecular docking studies should be supplemented with experimental work to confirm its efficacy

3.       Add literature support regarding the potential use of chlorogenic acid in the disease under consideration and provide a base for why this study was conducted

4.       Add appropriate references to the methods section

5.       Limitations of the current study should be added

6.       Carefully language edit the manuscript for typos and flow of information’s

 Comments on the Quality of English Language

English is otherwise fine, need careful check for typos and flow of information to be consistant from title till the discussion

Author Response

We sincerely thank the reviewer for your careful review and valuable comments. His observations and suggestions have been of great help in improving the quality and presentation of our manuscript. We have taken into account each of the points he mentioned.

  1. Authors need to establish the rationale for the use of chlorogenic acid in pulmonary arterial hypertension. Is it a random study or based on previous experimental results.

Answer: We thank you for your comment and your interest in the rationale for the use of CGA in the treatment of PAH in our study. We would like to provide a detailed explanation on the basis of our research and our rationale for exploring this molecule in the context of PAH.

Our decision to investigate CGA in PAH is based on the recent evaluation that our research team has conducted in in vitro and in vivo models of idiopathic pulmonary fibrosis (IPF), although these data have not yet been published. In these experiments, we observed positive effects of CGA in reducing the degree of fibrosis and lung inflammation, suggesting its therapeutic potential in the context of lung diseases.

Furthermore, PAH is considered to be, one of the serious complications associated with IPF, and given that CGA has shown benefits in pulmonary fibrosis, we were intrigued whether it could also have a positive impact on PAH. Surprisingly, we found limited information on the use of this molecule to treat PAH in the existing scientific literature.

Taking into account the above and the reviewer's observations, we have included in our introduction section works that support the use of CGA to treat some pulmonary diseases and specifically PAH, in order to improve the justification of our work, the following text is included in lines 63-88 of our manuscript, as shown below:

Recent research has suggested that CGA may have a potential therapeutic effect on various types of lung diseases including PAH [17-19]. It has been suggested that CGA prevents the oxidative, fibrotic and inflammatory effects of paraquat (PQ)-induced lung toxicity by enhancing antioxidant enzymes in murine models [17]. In addition, CGA was shown to reduce PQ-induced lung epithelial cell (AEC) apoptosis by preventing caspase 3 and caspase 9 cleavage and cytochrome c release from mitochondria to the cytoplasm, as well as reducing reactive oxygen species (ROS) production by increasing levels of the NF-E2-related factor 2 (Nrf2), superoxide dismutase 2 (SOD2) and glutathione [20]. Furthermore, CGA has been shown to inhibit the proliferation of human lung cancer A549 cell lines by inhibiting the binding of annexin A2 to the p50 subunit, resulting in the regulation of the expression of the antiapoptotic genes cIAP1 and cIAP2 of the NF-κB signaling pathway, resulting in a significant reduction in the proliferation of these tumor cell lines [18]. Additionally, available evidence suggests that CGA alleviates bleomycin-induced pulmonary fibrosis in mice by significantly improving lung inflammation and the degree of fibrosis through inhibition of endoplasmic reticulum stress, autophagy, and epithelial-mesenchymal transition (EMT) [19,21]. In relation to PAH, studies have suggested that CGA is capable of inhibiting hypoxia-induced pulmonary artery smooth muscle cell (PASMC) proliferation, one of the cellular processes closely related to vascular remodelling in PAH. These inhibitory effects were associated with reduction in Alpha hypoxia inducible factor 1 subunit alpha (HIF-1α) expression, G1 cell cycle arrest, and negative regulation of cell cycle proteins. Trials conducted in a murine model of monocrotaline-induced PAH in rats showed that CGA alleviates PAH by reducing intrapulmonary arterial hyperplasia [22]. However, despite the potential benefits of CGA in the treatment of lung diseases, such as PAH, the underlying molecular mechanism remains largely unexplored. Although beneficial effects of CGA have been observed, a deeper understanding of how this molecule exerts its positive effects is available.

  1. Network pharmacology and Molecular docking studies should be supplemented with experimental work to confirm its efficacy

Answer: We thank the reviewer for this important observation and suggestion regarding the need for experimental validation in our study. We want to explain our choice to focus on network pharmacology and molecular docking as the primary goals of this work.

First, the focus on network pharmacology allows us to identify potential interactions between CGA and PAH-related molecular targets at the level of biological systems. This analysis provides us with valuable insights into the key pathways and targets involved in the pathogenesis of PAH and how CGA might influence them. Furthermore, molecular docking provides us with a detailed understanding of the interactions at the molecular level between CGA and its therapeutic targets.

While we recognize the importance of experimental validation, it is important to highlight that network pharmacology and molecular docking are powerful tools that have proven to be instrumental in the early stage of research into new therapeutic agents. They allow us to perform an efficient initial screening based on biological information, which is crucial in the identification of possible candidates for future research.

It is important to highlight that, in the context of our research, experimental validation through in vivo and in vitro models presents certain difficulties and challenges that we have been addressing. Our research group is committed to conducting further work that addresses this need for experimental validation. While we recognize that experimental studies are essential to support our findings, we must overcome certain limitations and obstacles to conduct these experiments rigorously and effectively.

In our future efforts, we plan to undertake experimental research to validate the efficacy of CGA in relevant PAH models. These experiments will be carried out with the aim of consolidating our in-silico findings and providing a solid basis for the consideration of CGA as a potential therapeutic agent for PAH.

We believe that, given the circumstances and exploratory nature of this study, network pharmacology and molecular docking provide a solid and relevant foundation for our research objectives.

  1. Add literature support regarding the potential use of chlorogenic acid in the disease under consideration and provide a base for why this study was conducted

Answer: We appreciate your comment and your suggestion to add bibliographic support that justifies the potential use of CGA in PAH. We understand the importance of providing a solid foundation for our study and have been committed to improving the quality of our manuscript.

The choice to investigate CGA in the treatment of PAH is based on previous research and scientific evidence suggesting its therapeutic potential. We have identified several studies and scientific literature that support the use of CGA in various lung diseases, including those related to inflammation, pulmonary fibrosis, lung cancer and the regulation of cellular processes. However, we recognize the importance of incorporating these findings into the manuscript to provide a stronger basis for the rationale for our study.

In response to your suggestion, we have included the rationale for the use of CGA in the treatment of PAH according to the available evidence, this paragraph was included in the introduction section on lines 63-88 of our manuscript as shown below:

Recent research has suggested that CGA may have a potential therapeutic effect on various types of lung diseases including PAH [17-19]. It has been suggested that CGA prevents the oxidative, fibrotic and inflammatory effects of paraquat (PQ)-induced lung toxicity by enhancing antioxidant enzymes in murine models [17]. In addition, CGA was shown to reduce PQ-induced lung epithelial cell (AEC) apoptosis by preventing caspase 3 and caspase 9 cleavage and cytochrome c release from mitochondria to the cytoplasm, as well as reducing reactive oxygen species (ROS) production by increasing levels of the NF-E2-related factor 2 (Nrf2), superoxide dismutase 2 (SOD2) and glutathione [20]. Furthermore, CGA has been shown to inhibit the proliferation of human lung cancer A549 cell lines by inhibiting the binding of annexin A2 to the p50 subunit, resulting in the regulation of the expression of the antiapoptotic genes cIAP1 and cIAP2 of the NF-κB signaling pathway, resulting in a significant reduction in the proliferation of these tumor cell lines [18]. Additionally, available evidence suggests that CGA alleviates bleomycin-induced pulmonary fibrosis in mice by significantly improving lung inflammation and the degree of fibrosis through inhibition of endoplasmic reticulum stress, autophagy, and epithelial-mesenchymal transition (EMT) [19,21]. In relation to PAH, studies have suggested that CGA is capable of inhibiting hypoxia-induced pulmonary artery smooth muscle cell (PASMC) proliferation, one of the cellular processes closely related to vascular remodelling in PAH. These inhibitory effects were associated with reduction in Alpha hypoxia inducible factor 1 subunit alpha (HIF-1α) expression, G1 cell cycle arrest, and negative regulation of cell cycle proteins. Trials conducted in a murine model of monocrotaline-induced PAH in rats showed that CGA alleviates PAH by reducing intrapulmonary arterial hyperplasia [22]. However, despite the potential benefits of CGA in the treatment of lung diseases, such as PAH, the underlying molecular mechanism remains largely unexplored. Although beneficial effects of CGA have been observed, a deeper understanding of how this molecule exerts its positive effects is available.

  1. Add appropriate references to the methods section.

Answer: We appreciate your review and your valuable suggestions for improving our manuscript. We have taken your recommendation into account and have made the following modifications to the methods section:

We have added appropriate references in the methods section to better support and contextualize our procedures, approaches, software, and web-based tools used in our manuscript. These references will provide readers with a solid foundation for understanding the methodology we employed in our study.

  1. Limitations of the current study should be added

Answer: We welcome your review and suggestions. We recognize the importance of identifying and communicating the limitations of our study in order to provide a balanced and accurate view of the research conducted.

We have limitations that we consider pertinent to mention in the corresponding discussion section of the manuscript. Some of the limitations we have identified are included in lines 478-490 of our manuscript, as shown below:

In summary, our study presents promising findings and suggests that CGA may act on key targets and pathways associated with PAH, such as inflammation, vascular re-modelling, and hypoxia-related processes. These findings support the potential of CGA as a therapeutic agent for PAH by modulating these molecular targets. However, it is important to highlight some limitations inherent to this investigation. First, the association of some of the key targets with pathways such as lipid- and atherosclerosis-related pathways, which are not directly related to the pathophysiology of PAH, was observed. These observations underscore the need for further investigations to fully understand the scope of CGA action in PAH and to validate its potential efficacy in preclinical and clinical models. Furthermore, the study is based on in silico analysis and bioinformatics data, which requires experimental confirmation and further validation in in vitro and in vivo studies. Despite these limitations, the results of this research lay a solid foundation for future research in the field of PAH and new drug development.

  1. Carefully language edit the manuscript for typos and flow of information’s

Answer: We thank you for your review and your recommendations to improve the quality of the titled manuscript. Your observation about the need to carefully edit the language and detect typographical errors is of great importance to us.

We would like to emphasize that we have performed a thorough revision of the manuscript to address language and information flow issues. However, it is relevant to note that one of our collaborators, whose native language is English, has contributed significantly to this review. This collaborator has contributed its linguistic expertise to improve the coherence and clarity of the text, which has been of great help.

Round 2

Reviewer 3 Report

Comments and Suggestions for Authors

Authors have revised the manuscript. I have no further comments.